# (CERTIFIED!!) ADVERSARIAL ROBUSTNESS FOR FREE!

**Nicholas Carlini**[*1]  **Florian Tramèr**[*1]  **Krishnamurthy (Dj) Dvijotham**[1]
**Leslie Rice**[2]  **Mingjie Sun**[2]  **J. Zico Kolter**[2,3]
[1]Google  [2]Carnegie Mellon University  [3]Bosch Center for AI

## ABSTRACT

In this paper we show how to achieve state-of-the-art *certified* adversarial robustness to $\ell_2$-norm bounded perturbations by relying exclusively on off-the-shelf pretrained models. To do so, we instantiate the denoised smoothing approach of Salman et al. (2020) by combining a pretrained denoising diffusion probabilistic model and a standard high-accuracy classifier. This allows us to certify 71% accuracy on ImageNet under adversarial perturbations constrained to be within an $\ell_2$ norm of $\varepsilon = 0.5$, an improvement of 14 percentage points over the prior certified SoTA using any approach, or an improvement of 30 percentage points over denoised smoothing. We obtain these results using only pretrained diffusion models and image classifiers, without requiring any fine tuning or retraining of model parameters.

## 1  INTRODUCTION

Evaluating the robustness of deep learning models to norm bounded adversarial perturbations has been shown to be difficult (Athalye et al., 2018; Uesato et al., 2018). Certified defenses—such as those based on bound propagation (Gowal et al., 2018; Mirman et al., 2018) or randomized smoothing (Lecuyer et al., 2019; Cohen et al., 2019)—offer provable guarantees that a model's predictions are robust to norm-bounded adversarial perturbations, for a large fraction of examples in the test set.

The current state-of-the-art approaches to certify robustness to adversarial perturbations bounded in the $\ell_2$ norm rely on *randomized smoothing* (Lecuyer et al., 2019; Cohen et al., 2019). By taking a majority vote over the labels predicted by a "base classifier" under random Gaussian perturbations of the input, if the correct class is output sufficiently often, then the defense's output on the original un-noised input is guaranteed to be robust to $\ell_2$ norm bounded adversarial perturbations.

Denoised smoothing (Salman et al., 2020) is a certified defense that splits this one-step process into two. After randomly perturbing an input, the defense first applies a *denoiser* model that aims to remove the added noise, followed by a standard *classifier* that guesses a label given this *noised-then-denoised* input. This enables applying randomized smoothing to pretrained black-box base classifiers, as long as the denoiser can produce clean images close to the base classifier's original training distribution.

We observe that the recent line of work on *denoising diffusion probabilistic models* (Sohl-Dickstein et al., 2015; Ho et al., 2020; Nichol & Dhariwal, 2021)—which achieve state-of-the-art results on image generation—are a perfect match for the denoising step in a denoised smoothing defense. A forward diffusion process takes a source data distribution (e.g., images from some data distribution) and then adds Gaussian noise until the distribution converges to a high-variance isotropic Gaussian. Denoising diffusion models are trained to invert this process. Thus, we can use a diffusion model as a denoiser that recovers high quality denoised inputs from inputs perturbed with Gaussian noise.

In this paper, we combine state-of-the-art, publicly available diffusion models as denoisers with standard pretrained state-of-the-art classifiers. We show that the resulting denoised smoothing defense obtains significantly *better* certified robustness results—for perturbations of $\ell_2$ norm of $\epsilon \leq 2$ on ImageNet and $\epsilon \leq 0.5$ on CIFAR-10—compared to the "custom" denoisers trained in prior work (Salman et al., 2020), or in fact with any certifiably robust defense (even those that do not rely on denoised smoothing). Code to reproduce our experiments is available at: `https://github.com/ethz-privsec/diffusion_denoised_smoothing`.

---

[*]Joint first authors

## 2 BACKGROUND

**Adversarial examples** (Biggio et al., 2013; Szegedy et al., 2014) are inputs $x' = x + \delta$ constructed by taking some input $x$ (with true label $y \in \mathcal{Y}$) and adding a perturbation $\delta$ (that is assumed to be imperceptible and hence label-preserving) so that a given classifier $f$ misclassifies the perturbed input, i.e., $f(x + \delta) \neq y$. The "smallness" of $\delta$ is quantified by its Euclidean norm, and we constrain $\|\delta\|_2 \leq \varepsilon$. Even when considering exceptionally small perturbation budgets (e.g., $\varepsilon = 0.5$) modern classifiers often have near-0% accuracy (Carlini & Wagner, 2017).

**Randomized smoothing** (Lecuyer et al., 2019; Cohen et al., 2019) is a technique to certify the robustness of arbitrary classifiers against adversarial examples under the $\ell_2$ norm. Given an input $x$ and base classifier $f$, randomized smoothing considers a smooth version of $f$ defined as:

$$g(x) \coloneqq \operatorname{argmax}_c \Pr_{\delta \sim \mathcal{N}(0, \sigma^2 \mathbf{I})} (f(x + \delta) = c) \tag{1}$$

Cohen et al. (2019) prove that the smooth classifier $g$ is robust to perturbations of $\ell_2$ radius $R$, where the radius $R$ grows with the classifier's "margin" (i.e., the difference in probabilities assigned to the most likely and second most-likely classes). As the probability in Equation 1 cannot be efficiently computed when the base classifier $f$ is a neural network, Cohen et al. (2019) instantiate this defense by sampling a small number $m$ of noise instances (e.g., $m = 10$) and taking a majority vote over the outputs of the base classifier $f$ on $m$ noisy versions of the input. To compute a lower-bound on this defense's robust radius $R$, they estimate the probabilities $\Pr[f(x + \delta) = c]$ for each class label $c$ by sampling a large number $N$ of noise instances $\delta$ (e.g., $N = 100{,}000$). See Cohen et al. (2019) for details.

**Denoised smoothing** (Salman et al., 2020) is an instantiation of randomized smoothing, where the base classifier $f$ is composed of a denoiser `denoise` followed by a standard classifier $f_{\texttt{clf}}$:

$$f(x + \delta) \coloneqq f_{\texttt{clf}}(\texttt{denoise}(x + \delta)) . \tag{2}$$

Given a very good denoiser (i.e., $\texttt{denoise}(x + \delta) \approx x$ with high probability for $\delta \sim \mathcal{N}(0, \sigma^2 \mathbf{I})$), we can expect the base classifier's accuracy on noisy images to be similar to the *clean* accuracy of the standard classifier $f_{\texttt{clf}}$. Salman et al. (2020) instantiate their denoised smoothing technique by training custom denoiser models with Gaussian noise augmentation, combined with off-the-shelf pretrained classifiers.

**Denoising Diffusion Probabilistic Models** (Sohl-Dickstein et al., 2015; Ho et al., 2020; Nichol & Dhariwal, 2021) are a form of generative model that work by learning a model that can reverse time on a diffusion process of the form $x_t \sim \sqrt{1 - \beta_t} \cdot x_{t-1} + \beta_t \cdot \omega_t, \omega_t \sim \mathcal{N}(0, \mathbf{I})$ with $x_0$ coming from the data distribution, and the $\beta_t$ being fixed (or learned) variance parameters. The diffusion process transforms images from the target data distribution to purely random noise over time. The reverse process then synthesizes images from the data distribution starting with random Gaussian noise. In this paper we will not make use of diffusion models in the typical way; instead it suffices to understand just one single property about how they are trained.

Given a clean training image $x \in [-1, 1]^{w \cdot h \cdot c}$, a diffusion model selects a *timestep* $t \in \mathbb{N}^+$ from some fixed schedule and then samples a noisy image $x_t$ of the form

$$x_t \coloneqq \sqrt{\alpha_t} \cdot x + \sqrt{1 - \alpha_t} \cdot \mathcal{N}(0, \mathbf{I}) , \tag{3}$$

where the factor $\alpha_t$ is a constant derived from the timestamp $t$ that determines the amount of noise to be added to the image (the noise magnitude increases monotonically with $t$).

The diffusion model is then trained (loosely speaking) to minimize the discrepancy between $x$ and $\texttt{denoise}(x_t; t)$; that is, to predict what the original (un-noised) image should look like after applying the noising step at timestep $t$.[1]

---

[1] State-of-the-art diffusion models are actually trained to predict the *noise* rather than the denoised image directly (Ho et al., 2020; Nichol & Dhariwal, 2021).

| **Algorithm 1** Noise, denoise, classify | **Algorithm 2** Randomized smoothing (Cohen et al., 2019) |
|---|---|
| 1: NOISEANDCLASSIFY$(x, \sigma)$: | 1: PREDICT$(x, \sigma, N, \eta)$: |
| 2: $\quad t^\star, \alpha_{t^\star} \leftarrow$ GETTIMESTEP$(\sigma)$ | 2: $\quad$ counts $\leftarrow \mathbf{0}$ |
| 3: $\quad x_{t^\star} \leftarrow \sqrt{\alpha_{t^\star}}(x + \mathcal{N}(0, \sigma^2 \mathbf{I}))$ | 3: $\quad$ **for** $i \in \{1, 2, \ldots, N\}$ **do** |
| 4: $\quad \hat{x} \leftarrow$ denoise$(x_{t^\star}; t^\star)$ | 4: $\quad\quad y \leftarrow$ NOISEANDCLASSIFY$(x, \sigma)$ |
| 5: $\quad y \leftarrow f_{\text{clf}}(\hat{x})$ | 5: $\quad\quad$ counts$[y] \leftarrow$ counts$[y] + 1$ |
| 6: $\quad$ **return** $y$ | 6: $\quad \hat{y}_A, \hat{y}_B \leftarrow$ top two labels in counts |
| 7: | 7: $\quad n_A, n_B \leftarrow$ counts$[\hat{y}_A]$, counts$[\hat{y}_B]$ |
| 8: GETTIMESTEP$(\sigma)$: | 8: $\quad$ **if** BINOMPTEST$(n_A, n_A + n_B, 1/2) \leq \eta$ **then** |
| 9: $\quad t^\star \leftarrow$ find $t$ s.t. $\frac{1 - \alpha_t}{\alpha_t} = \sigma^2$ | 9: $\quad\quad$ **return** $\hat{y}_A$ |
| 10: $\quad$ **return** $t^\star, \alpha_{t^\star}$ | 10: $\quad$ **else** |
| | 11: $\quad\quad$ **return** Abstain |

Figure 1: Our approach can be implemented in under 15 lines of code, given an off-the-shelf classifier $f_{\text{clf}}$ and an off-the-shelf diffusion model denoise. The PREDICT function is adapted from Cohen et al. (2019) and takes as input a number of noise samples $N$ and a statistical significance level $\eta \in (0, 1)$ and inherits the same robustness certificate proved in Cohen et al. (2019).

## 3 DIFFUSION DENOISED SMOOTHING

Our approach, Diffusion Denoised Smoothing (DDS), requires no new technical ideas on top of what was introduced in the section above.

**Denoised smoothing via a diffusion model.** The only minor technicality required for our method is to map between the noise model required by randomized smoothing and the noise model used within diffusion models. Specifically, randomized smoothing requires a data point augmented with additive Gaussian noise $x_{\text{rs}} \sim \mathcal{N}(x, \sigma^2 \mathbf{I})$, whereas diffusion models assume the noise model $x_t \sim \mathcal{N}(\sqrt{\alpha_t}x, (1 - \alpha_t)\mathbf{I})$. Scaling $x_{\text{rs}}$ by $\sqrt{\alpha_t}$ and equating the variances yields the relationship

$$\sigma^2 = \frac{1 - \alpha_t}{\alpha_t} \, . \tag{4}$$

Thus, in order to employ a diffusion model for randomized smoothing at a given noise level $\sigma$, we first find the timestep $t^\star$ such that $\sigma^2 = \frac{1 - \alpha_{t^\star}}{\alpha_{t^\star}}$; the precise formula for this equation will depend on the schedule of the $\alpha_t$ terms used by the diffusion model, but this can typically be computed in closed form, even for reasonably complex diffusion schedules.[2] Next, we compute

$$x_{t^\star} = \sqrt{\alpha_{t^\star}}(x + \delta), \;\; \delta \sim \mathcal{N}(0, \sigma^2 \mathbf{I}) \tag{6}$$

and apply the diffusion denoiser on $x_{t^\star}$ to obtain an estimate of the denoised sample

$$\hat{x} = \text{denoise}(x_{t^\star}; t^\star) \, . \tag{7}$$

And finally, we classify the estimated denoised image with an off-the-shelf classifier

$$y = f_{\text{clf}}(\hat{x}) \, . \tag{8}$$

The entirety of this algorithmic approach is shown in Figure 1.

---

[2]For example, in Nichol & Dhariwal (2021), the authors advocate for the schedule $\alpha_t = f(t)/f(0)$, where $f(t) = \cos\left(\frac{t/T + s}{1 + s} \cdot \frac{\pi}{2}\right)^2$ for various values of $T$, and $s$ discussed in this reference. In this case, for a given desired value of $\sigma^2$, some algebra yields the solution for $t$

$$t^\star = T\left(1 - \frac{2(1 + s)\csc^{-1}\left(\sqrt{1 + \sigma^2}\csc\left(\frac{\pi}{2 + 2s}\right)\right)}{\pi}\right) \, . \tag{5}$$

The actual formula here is unimportant and only shown as an illustration of how such computation can look in practice. Even when such a closed form solution does not exist, because the schedules for $\alpha_t$ are monotonic decreasing, one can always find a solution via 1D root-finding methods if necessary.

To obtain a robustness certificate, we repeat the above denoising process many times (e.g., 100,000) and compute the certification radius using the approach of Cohen et al. (2019) (note that since our diffusion models expects inputs in $[-1, 1]^d$, we then divide the certified radius by 2 to obtain a certified radius for inputs in $[0, 1]$ as assumed in all prior work).

**One-shot denoising.** Readers familiar with diffusion models may recall that the standard process repeatedly applies a "single-step" denoising operation $x_{t-1} = d(x_t; t)$ that aims to convert a noisy image at some timestep $t$ to a (slightly less) noisy image at the previous timestep $t - 1$. The full diffusion process would then be defined by the following iterative procedure:

$$\tilde{x} = \texttt{denoise}_{\text{iter}}(x + \delta; t) \coloneqq d(d(\ldots d(d(x + \delta; t); t - 1) \ldots; 2); 1) \ .$$

In fact, each application of the one-step denoiser $d$ consists of two steps: (1) an estimation of the fully denoised image $x$ from the current timestep $t$, and (2) computing a (properly weighted, according to the diffusion model) average between this estimated denoised image and the noisy image at the previous timestep $t - 1$. Thus, instead of performing the entire $t$-step diffusion process to denoise an image, it is also possible to run the diffusion step $d$ *once* and simply output the best estimate for the denoised image $x$ in one shot.

When a diffusion model generates images from scratch (i.e., the denoiser is applied to pure noise), the iterative process gives higher fidelity outputs than this one-shot approach (Ho et al., 2020). But here, where we aim to denoise one particular image, a one-shot approach has two advantages:

1. **High accuracy**: it turns out that standard pretrained classifiers are more accurate on one-shot denoised images compared to images denoised with the full $t$-steps of denoising. We hypothesize this is due to the fact that when we first apply the single-step denoiser $d$ at timestep $t$, the denoiser already has all the available information about $x$. By applying the denoiser multiple times, we can only *destroy* information about $x$ as each step adds new (slightly smaller) Gaussian noise. In fact, by using the iterative $t$-step denoising strategy, we are in essence pushing part of the classification task onto the denoiser, in order to decide how to fill in the image. Section 5 experimentally validates this hypothesis.

2. **Improved efficiency**: instead of requiring several hundred (or thousand) forward passes to denoise any given image, we only require one single pass. This is especially important when we perform many thousand predictions as is required for randomized smoothing to obtain a robustness certificate.

**Related work.** We are not the first to observe a connection between randomized smoothing and diffusion models. The work of Lee (2021) first studied this problem—however they do not obtain significant accuracy improvements, likely due to the fact that diffusion models available at the time that work was done were not good enough. Separately, Nie et al. (2022) suggest that diffusion models might be able to provide strong *empirical* robustness to adversarial examples, as evaluated by robustness under adversarial attacks computed using existing attack algorithms; this is orthogonal to our results.

## 4 EVALUATION

We evaluate diffusion denoised smoothing on two standard datasets, CIFAR-10 and ImageNet, and find it gives state-of-the-art certified $\ell_2$ robustness on both. On CIFAR-10, we draw $N = 100{,}000$ noise samples and on ImageNet we draw $N = 10{,}000$ samples to certify the robustness following Cohen et al. (2019).

As is standard in prior work, we perform randomized smoothing for three different noise magnitudes, $\sigma \in \{0.25, 0.5, 1.0\}$. For a fair comparison to prior work in Table 2 and Table 1, we give the best results reported in each paper across these same three noise magnitudes. Note that prior work only uses three levels of noise due to the computational overhead; one benefit of using a diffusion model is we could have used other amounts of noise without training a new denoiser model.

**CIFAR-10 configuration.** We denoise CIFAR-10 images with the 50M-parameter diffusion model from Nichol & Dhariwal (2021).[3] The denoised images are classified with a 87M-parameter

---

[3] https://github.com/openai/improved-diffusion

| Method | Off-the-shelf | Extra data | Certified Accuracy at $\varepsilon$ (%) | | | | |
|---|---|---|---|---|---|---|---|
| | | | 0.5 | 1.0 | 1.5 | 2.0 | 3.0 |
| PixelDP (Lecuyer et al., 2019) | ○ | ✗ | (33.0)16.0 | - | - | | |
| RS (Cohen et al., 2019) | ○ | ✗ | (67.0)49.0 | (57.0)37.0 | (57.0)29.0 | (44.0)19.0 | (44.0)12.0 |
| SmoothAdv (Salman et al., 2019) | ○ | ✗ | (65.0)56.0 | (54.0)43.0 | (54.0)37.0 | (40.0)27.0 | (40.0)20.0 |
| Consistency (Jeong & Shin, 2020) | ○ | ✗ | (55.0)50.0 | (55.0)44.0 | (55.0)34.0 | (41.0)24.0 | (41.0)17.0 |
| MACER (Zhai et al., 2020) | ○ | ✗ | (68.0)57.0 | (64.0)43.0 | (64.0)31.0 | (48.0)25.0 | (48.0)14.0 |
| Boosting (Horváth et al., 2022a) | ○ | ✗ | (65.6)57.0 | (57.0)44.6 | (57.0)**38.4** | (44.6)**28.6** | (38.6)**21.2** |
| DRT (Yang et al., 2021) | ○ | ✗ | (52.2)46.8 | (55.2)44.4 | (49.8)**39.8** | (49.8)**30.4** | (49.8)**23.4** |
| SmoothMix (Jeong et al., 2021) | ○ | ✗ | (55.0)50.0 | (55.0)43.0 | (55.0)**38.0** | (40.0)26.0 | (40.0)20.0 |
| ACES (Horváth et al., 2022b) | ◐ | ✗ | (63.8)54.0 | (57.2)42.2 | (55.6)35.6 | (39.8)25.6 | (44.0)19.8 |
| Denoised (Salman et al., 2020) | ◐ | ✗ | (60.0)33.0 | (38.0)14.0 | (38.0)6.0 | - | - |
| Lee (Lee, 2021) | ● | ✗ | 41.0 | 24.0 | 11.0 | - | - |
| **Ours** | ● | ✓ | (82.8)**71.1** | (77.1)**54.3** | (77.1)**38.1** | (60.0)**29.5** | (60.0)13.1 |

Table 1: ImageNet certified top-1 accuracy for prior defenses on randomized smoothing and denoised smoothing. Randomized smoothing techniques rely on special-purpose models (indicated by a empty circle). The work of Horváth et al. (2022b) is an exception in that it selectively applies either a robust or accurate off-the-shelf classifier (indicated by a half full circle). Denoised smoothing (Salman et al., 2020) use an off-the-shelf classifier but train their own denoiser (indicated by a half full circle). Our base approach uses an off-the-shelf classifier and off-the-shelf denoiser (indicated by a full circle). Each entry lists the certified accuracy, with the clean accuracy for that model in parentheses, using numbers taken from respective papers.

ViT-B/16 model (Dosovitskiy et al., 2021) that was pretrained on ImageNet-21k (Deng et al., 2009) (in $224 \times 224$ resolution) and finetuned on CIFAR-10. We use the implementation from HuggingFace[4] which reaches $97.9\%$ test accuracy on CIFAR-10. In addition, we also report results with a standard 36M parameter Wide-ResNet-28-10 model (Zagoruyko & Komodakis, 2016) trained on CIFAR-10 to 95.2% accuracy.

As is typical, we report results with images normalized to $[0, 1]^{32 \times 32 \times 3}$. We obtain a throughput of 825 images per second through the diffusion model and ViT classifier on an A100 GPU at a batch size of 1,000. We report robust accuracy results averaged over the entire CIFAR-10 test set.

**ImageNet configuration.** We denoise ImageNet images with the 552M-parameter class-unconditional diffusion model from Dhariwal & Nichol (2021), and classify images with the 305M-parameter BEiT large model (Bao et al., 2022) which reaches a $88.6\%$ top-1 validation accuracy using the implementation from `timm` (Wightman, 2019). We report results for our images when normalized to $[0, 1]^{224 \times 224 \times 3}$ to allow us to compare to prior work. The overall latency of this joint denoise-then-classify model is 1.5 seconds per image on an A100 GPU at a batch size of 32. We report results averaged over 1,000 images randomly selected from the ImageNet test set.

## 4.1 RESULTS

On both CIFAR-10 and ImageNet we outperform the state-of-the-art denoised smoothing approaches (i.e., Salman et al. (2020) and Lee (2021)) in every setting; see Table 1 and Table 2, as well as Figure 2 for detailed results. Perhaps even more impressively, we also outperform models trained with randomized smoothing at low $\varepsilon$ distortions ($\epsilon \le 0.5$ on CIFAR-10, and $\epsilon \le 2$ on ImageNet), and nearly match them at high $\varepsilon$. Even though these randomized smoothing techniques train their models end-to-end and specifically design these models to have high accuracy on Gaussian noise, we find that our approach's use of off-the-shelf models yields superior robustness (and much higher clean accuracy as an added bonus).

Interestingly, we find that using a diffusion model to perform the denoising step gives its most significant benefits when $\sigma$ and $\varepsilon$ are small: for example, while we reach 71.1% top-1 accuracy at $\varepsilon = 0.5$ on ImageNet, an improvement over prior work of $+14$ percentage points, when we reach $\varepsilon = 3$ our scheme is 7 percentage points worse than state-of-the-art. Our hypothesis for this effect,

---

[4] https://huggingface.co/aaraki/vit-base-patch16-224-in21k-finetuned-cifar10

| Method | Off-the-shelf | Extra data | Certified Accuracy at $\varepsilon$ (%) | | | |
|---|---|---|---|---|---|---|
| | | | 0.25 | 0.5 | 0.75 | 1.0 |
| PixelDP (Lecuyer et al., 2019) | ○ | ✗ | (71.0)22.0 | (44.0)2.0 | - | - |
| RS (Cohen et al., 2019) | ○ | ✗ | (75.0)61.0 | (75.0)43.0 | (65.0)32.0 | (66.0)22.0 |
| SmoothAdv (Salman et al., 2019) | ○ | ✗ | (75.6)67.4 | (75.6)57.6 | (74.8)47.8 | (57.4)38.3 |
| SmoothAdv (Salman et al., 2019) | ○ | ✓ | (84.3)74.9 | (80.1)63.4 | (80.1)**51.9** | (62.2)**39.6** |
| Consistency (Jeong & Shin, 2020) | ○ | ✗ | (77.8)68.8 | (75.8)58.1 | (72.9)48.5 | (52.3)37.8 |
| MACER (Zhai et al., 2020) | ○ | ✗ | (81.0)71.0 | (81.0)59.0 | (66.0)46.0 | (66.0)38.0 |
| Boosting (Horváth et al., 2022a) | ○ | ✗ | (83.4)70.6 | (76.8)60.4 | (71.6)**52.4** | (52.4)**38.8** |
| DRT (Yang et al., 2021) | ○ | ✗ | (81.5)70.4 | (72.6)60.2 | (71.9)50.5 | (56.1)**39.8** |
| SmoothMix (Jeong et al., 2021) | ○ | ✗ | (77.1)67.9 | (77.1)57.9 | (74.2)47.7 | (61.8)37.2 |
| ACES (Horváth et al., 2022b) | ◑ | ✗ | (79.0)69.0 | (74.2)57.2 | (74.2)47.0 | (58.6)37.8 |
| Denoised (Salman et al., 2020) | ◑ | ✗ | (72.0)56.0 | (62.0)41.0 | (62.0)28.0 | (44.0)19.0 |
| Lee (Lee, 2021) | ● | ✗ | 60.0 | 42.0 | 28.0 | 19.0 |
| **Ours** | ● | ✓ | (88.1)76.7 | (88.1)63.0 | (88.1)45.3 | (77.0)32.1 |
| **Ours (+finetuning)** | ◑ | ✓ | (91.2)**79.3** | (91.2)**65.5** | (87.3)48.7 | (81.5)35.5 |

Table 2: CIFAR-10 certified accuracy for prior defenses from the literature. The columns have the same meaning as in Table 1.

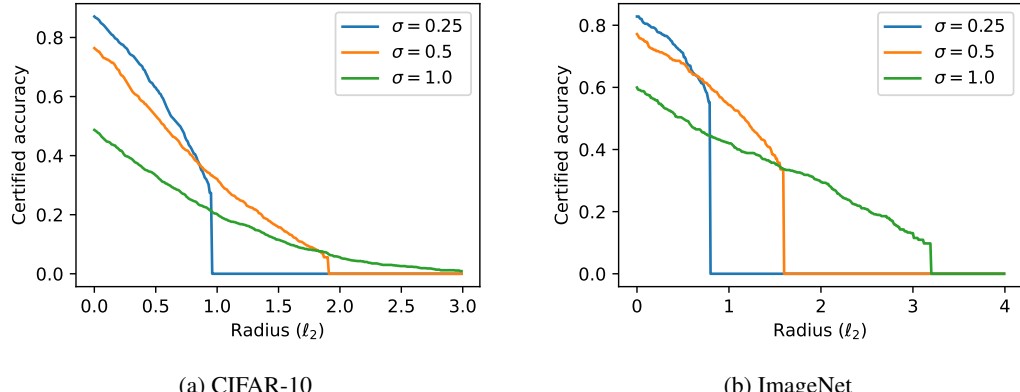

(a) CIFAR-10           (b) ImageNet

Figure 2: Certified accuracy as a function of the $\ell_2$ adversarial perturbation bound, when varying levels of Gaussian noise $\sigma \in \{0.25, 0.5, 1.0\}$. Bounds are computed with 100,000 samples per run on CIFAR-10, and 10,000 on ImageNet.

which we explore further in Section 5, is that diffusion models are prone to "hallucinate" content when denoising extremely noisy images. Thus, instead of reinforcing the signal from the correct class, the diffusion model generates a signal from another class, thereby fooling the classifier.

**CIFAR-10 ablation.** The off-the-shelf classifiers we use were pretrained on larger datasets than respectively CIFAR-10 and ImageNet. It is well known that the use of additional data can boost robustness, both for empirical (Schmidt et al., 2018) and certified (Salman et al., 2019) defenses. To investigate the role played by the pretrained model, we repeat our CIFAR-10 experiment using a standard Wide-ResNet-28-10 model (Zagoruyko & Komodakis, 2016) trained solely on CIFAR-10 to 95.2% accuracy. The results with this classifier (see Table 6a) outperform prior denoised smoothing approaches, and are competitive with prior randomized smoothing results up to $\epsilon = 0.5$.

The ViT classifier outperforms the ResNet because it is more robust to the distribution shift introduced by the noising-and-denoising procedure. To alleviate this, we can further *finetune* the classifier on denoised images denoise$(x + \delta)$ from the CIFAR-10 training set. This defense is thus not strictly "off-the-shelf" anymore (although finetuning is negligible compared to the training time of the diffusion model and classifier). Table 6b shows that a finetuned Wide-ResNet achieves comparable-or-better results than a non-finetuned ViT. Thus, with a minimal amount of training, we also surpass prior randomized smoothing results *without relying on any external data*. If we finetune

| Method | Off-the-shelf | Extra data | Certified Accuracy at $\varepsilon$ (%) | | | |
|---|---|---|---|---|---|---|
| | | | 0.25 | 0.5 | 0.75 | 1.0 |
| Wide-ResNet | ● | ✗ | $^{(83.8)}$70.6 | $^{(83.8)}$55.7 | $^{(83.8)}$40.0 | $^{(65.8)}$26.1 |
| ViT | ● | ✓ | $^{(88.1)}$76.7 | $^{(88.1)}$63.0 | $^{(88.1)}$45.3 | $^{(77.0)}$32.1 |
| Wide-ResNet +finetune | ◑ | ✗ | $^{(85.9)}$76.7 | $^{(85.9)}$**63.8** | $^{(85.9)}$**49.5** | $^{(74.5)}$**36.4** |
| ViT +finetune | ◑ | ✓ | $^{(91.2)}$**79.3** | $^{(91.2)}$**65.5** | $^{(91.2)}$48.7 | $^{(81.5)}$**35.5** |

Table 3: Summary of our ablation on CIFAR-10. The diffusion model and Wide-ResNet classifier are trained solely on CIFAR-10, while the ViT classifier is pretrained on a larger dataset. The finetuning results are obtained by taking an off-the-shelf diffusion model and classifier, and tuning the classifier on noised-then-denoised images from CIFAR-10.

the ViT model (Table 6d), we further improve our defense's clean accuracy and certified robustness at $\epsilon \leq 0.5$ by a couple of percentage points. Our ablation is summarized in Table 3.

## 5 ANALYSIS AND DISCUSSION

We achieve state-of-the-art certified accuracy using diffusion models despite the fact that we are not using these models as *diffusion models* but rather *trivial denoisers*. That is, instead of leveraging the fact that diffusion models can *iteratively* refine images across a *range of noise levels*, we simply apply the diffusion model *once* for a *fixed* noise level, to perform one-shot denoising.

In this section we study why this approach outperforms prior work that trained straightforward denoisiers for denoised smoothing (Salman et al., 2020), and why using diffusion models for one-shot denoising performs better than the more involved iterative diffusion process. Last we show promising results of multi-step diffusion using an advanced deterministic sampler.

### 5.1 FULL DIFFUSION VERSUS ONE-SHOT DENOISING

When used as generative models, diffusion models perform denoising through an iterative process that repeatedly refines an estimate of the final denoised image. When given an image $x_t$ with noise of magnitude corresponding to some diffusion timestep $t$, the model first predicts a one-shot estimate of the denoised image $x_0$, and then constructs an estimate $x_{t-1}$ of the noised image at timestep $t - 1$ by interpolating (with appropriate weights) between $x_0$, $x_t$ *and fresh isotropic Gaussian noise* $\mathcal{N}(0, I)$. The diffusion process is then applied recursively at timestep $t - 1$.

Intuitively, it may be expected that when using a diffusion model as a denoiser, one-shot denoising will produce more faithful results than the full iterative reverse-diffusion process. Indeed, each step of the reverse-diffusion process *destroys information* about the original image, since each step adds fresh Gaussian noise to the image. Thus, information theoretically at least, it should be easier to denoise an image in one-shot than over multiple iterations.

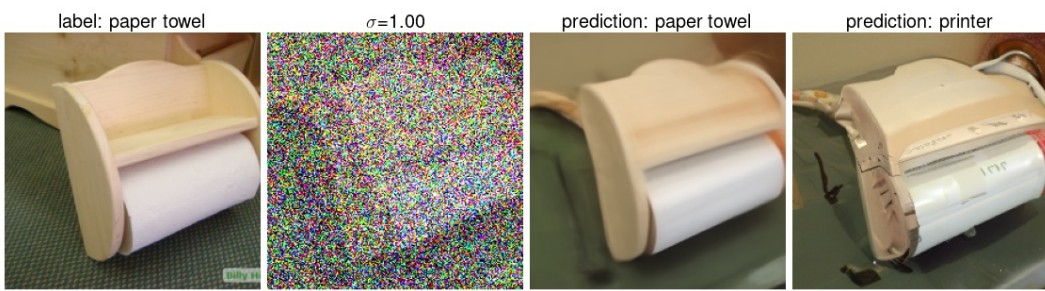

Figure 3: Intuitive examples for why multi-step denoised images are less recognized by the classifier. From left to right: clean images, noisy images with $\sigma = 1.0$, one-step denoised images, multi-step denoised images. For the denoised images, we show the prediction by the pretrained BEiT model.

We find that this is indeed the case. While the full reverse-diffusion process produces denoised images with more finegrained details (which is a good property for generating photorealistic images from scratch), these details are often not actually faithful to the original image we want to denoise. Instead, diffusion models are prone to "hallucinate" salient detailed features during the iterative denoise-and-noise process. We illustrate some examples of this hallucination phenomenon in Figure 3. Here, we noise an original image (on the left) with large Gaussian noise ($\sigma = 1$) and then apply either the full reverse-diffusion process (rightmost image) or a one-shot denoising at the appropriate timestep (2nd image to the right). As we can see, one-shot denoising produces mostly faithful, but blurry, reconstructions of the original image, with finegrained details lost due to noise. In contrast, iterative denoising "invents" new details that result in images that are ultimately more photorealistic but semantically different from the starting image. Additional examples (with multiple random seeds) are in Figure 4 and Figure 5 in the Appendix.

## 5.2 TRAINING ON RESTRICTED NOISE LEVELS

Given that one-shot denoising performs better than full multi-shot denoising, we now turn to understanding our next question: if we are just using diffusion models as one-shot denoisers, then why do diffusion models perform better compared to the straightforward denoisers trained in prior work (Salman et al., 2020)? To investigate this, we train seven new diffusion models on CIFAR-10 with varying levels of Gaussian noise—all the way towards a model trained on a single noise level, i.e., a straightforward denoiser.

Recall that during standard training of a diffusion model, we sample a timestep $T$ uniformly from some range, add noise according to this timestep, and then train the model to predict the noise that has been added. The only difference between this process and the standard denoised smoothing training process (Salman et al., 2020) is the fact that here we are training on multiple levels of Gaussian noise simultaneously. Therefore we now perform a comparative analysis of models trained on more restrictive noise levels. We select seven different levels of noise:

- Three models are trained exclusively on Gaussian noise of fixed standard deviation of respectively $\sigma = 0.25$, $\sigma = 0.5$, or $\sigma = 1.0$. This is identical to training a "straightforward" denoiser on noise of a fixed magnitude.
- One model is trained on all three noise levels at the same time.
- Two models are trained on noise uniformly selected from $\sigma \in [0, 0.25]$, and $\sigma \in [0, 1.0]$.
- One model is trained using the full range of noise, from $\sigma \in [0, S]$ for some $S \gg 1$ (the exact value of $S$ depends on the chosen noise schedule for the diffusion model).

We then evaluate the clean accuracy of an off-the-shelf ViT model on each image when denoised (in one shot) with each of these diffusion models, where the images are noised with a standard deviation of either $\sigma = 0.25$, $\sigma = 0.5$, or $\sigma = 1.0$. The results are summarized in Table 4.

| | Noise at evaluation | | |
|---|---|---|---|
| Training noise | $\sigma = 0.25$ | $\sigma = 0.5$ | $\sigma = 1.0$ |
| $\sigma \in \{0.25\}$ | 79.0 | 16.2 | 9.8 |
| $\sigma \in \{0.5\}$ | 14.5 | 60.1 | 15.4 |
| $\sigma \in \{1.0\}$ | 13.9 | 13.5 | 35.5 |
| $\sigma \in \{0.25, 0.5, 1.0\}$ | 81.6 | 68.1 | 43.0 |
| $\sigma \in [0, 0.25]$ | 84.5 | 14.5 | 9.9 |
| $\sigma \in [0, 1.0]$ | 84.0 | 71.6 | **46.0** |
| $\sigma \in [0, S \gg 1]$ (standard) | **85.5** | **72.3** | 44.8 |

Table 4: Clean accuracy of an off-the-shelf ViT classifier on images denoised with a diffusion model trained on restricted levels of Gaussian noise. Diffusion models trained on more diverse noise ranges yield higher accuracy on one-shot denoised images, even compared to models trained on the specific noise level used at evaluation time.

As expected, training a new model on any one individual noise level, and then using that model to denoise images at that noise level, gives high downstream accuracy: for example, training a diffusion

model using $\sigma = 0.25$ noise and then evaluating at this same noise level gives 79% accuracy. However if we then try and use this model to denoise images at a different noise level—say $\sigma = 0.5$—the accuracy of the classifier drops to just 16%. If we train the diffusion model directly on $\sigma = 0.5$ noise, we instead get a much better classification accuracy of 60.1%, but without good generalization to lower or higher noise levels. Similarly, training on noise of $\sigma = 1.0$ only gives good results when denoising images with the same noise level.

More surprisingly, however, is that training on all three noise levels *simultaneously* gives better accuracy for denoising images at each noise level, compared to a diffusion model trained specifically and solely for that noise level. For example, when denoising images with $\sigma = 0.5$ Gaussian noise, we get a classification accuracy of 68.1% when the diffusion model is trained on that noise level *and* additional lower and higher noise levels—a value 8% higher than the accuracy of 60.1% we get when training the diffusion model solely on $\sigma = 0.5$ noise.

If we train on more granular noise levels, either in $[0, 0.25]$ or in the full interval $[0, 1]$, the classification accuracy on denoised images at the three individual noise levels further increases by a few percentage points. Quite surprisingly, the standard training regime which trains the diffusion model on noise from a larger range $[0, S]$ for some $S \gg 1$ further improves the denoising capabilities at low noise levels ($\sigma = 0.25$ and $\sigma = 0.5$), but slightly harms the accuracy for larger noise ($\sigma = 1.0$).

From this experiment, we can conclude that the (full) training process of diffusion models leads to much better, and more generalizable, one-shot denoising capabilities than when training a standalone denoiser on a single noise level as in prior work.

## 5.3 ADVANCED DETERMINISTIC MULTI-STEP SAMPLER

In section 5.1, we found that the denoised images from full multi-step diffusion have a tendency to deviate from the original clean image. This could be due to the stochastic nature of the full reverse-diffusion process, since at each step a random noise is added. We notice a line of work (Song et al., 2021; Karras et al., 2022) on fast deterministic sampling of diffusion models. We show that with such an advanced sampler, multi-step diffusion is able to beat one-shot denoising.

We consider the deterministic EDM sampler proposed by Karras et al. (2022). We compare the recognizability of images denoised by EDM sampler and one-shot denoising. We adapt EDM sampler for image denoising by setting the maximum noise sigma of the sampling noise schedule to be the noise level found by Equation 4. We use the suggested sampler setting from Karras et al. (2022) on CIFAR-10, where 18 reverse steps with 35 evaluations of the diffusion model are performed for each example. The result is summarized in Table 5. We can see that the deterministic EDM sampler is superior over one-shot denoising.

| Classifier | Method | $\sigma = 0.25$ | $\sigma = 0.5$ | $\sigma = 1.0$ |
|---|---|---|---|---|
| Wide-ResNet | One-shot denoising | 81.3 | 64.0 | 35.8 |
| | EDM sampler | 85.0 | 73.0 | 53.8 |
| VIT | One-shot denoising | 84.9 | 71.6 | 50.8 |
| | EDM sampler | 86.1 | 73.1 | 54.0 |

Table 5: Clean accuracy (average over 5 runs) of off-the-shelf CIFAR-10 classifiers evaluated on images denoised by one-shot denoising and EDM sampler (Karras et al., 2022).

## 6 CONCLUSION

At present, training certified adversarially robust deep learning models requires specialized techniques explicitly designed for the purpose of performing provably robust classification (Cohen et al., 2019). While this has proven effective, these models are extremely difficult to train to high accuracy, and degrade clean accuracy significantly.

We suggest an alternative approach is possible. By exclusively making use of off-the-shelf models designed to be state-of-the-art at classification and image denoising, we can leverage the vast resources dedicated to training highly capable models for the new purpose of robust classification.

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

# A   APPENDIX

| | Certified Accuracy at $\varepsilon$ (%) | | | | |
|---|---|---|---|---|---|
| Noise | 0.0 | 0.25 | 0.5 | 0.75 | 1.0 |
| $\sigma = 0.25$ | 83.8 | 70.6 | 55.7 | 40.0 | 0.0 |
| $\sigma = 0.5$ | 65.8 | 54.7 | 43.7 | 34.2 | 26.1 |
| $\sigma = 1.0$ | 33.2 | 28.0 | 22.8 | 18.0 | 13.6 |

(a) Wide-ResNet

| | Certified Accuracy at $\varepsilon$ (%) | | | | |
|---|---|---|---|---|---|
| Noise | 0.0 | 0.25 | 0.5 | 0.75 | 1.0 |
| $\sigma = 0.25$ | 85.9 | 76.7 | 63.8 | **49.5** | 0.0 |
| $\sigma = 0.5$ | 74.5 | 66.0 | 56.1 | 45.7 | **36.4** |
| $\sigma = 1.0$ | 55.1 | 48.7 | 42.3 | 35.8 | 29.9 |

(b) Finetuned Wide-ResNet

| | Certified Accuracy at $\varepsilon$ (%) | | | | |
|---|---|---|---|---|---|
| Noise | 0.0 | 0.25 | 0.5 | 0.75 | 1.0 |
| $\sigma = 0.25$ | 88.1 | 76.7 | 63.0 | 45.3 | 0.0 |
| $\sigma = 0.5$ | 77.0 | 65.8 | 53.4 | 41.8 | 32.1 |
| $\sigma = 1.0$ | 49.5 | 40.3 | 33.3 | 26.1 | 20.2 |

(c) ViT

| | Certified Accuracy at $\varepsilon$ (%) | | | | |
|---|---|---|---|---|---|
| Noise | 0.0 | 0.25 | 0.5 | 0.75 | 1.0 |
| $\sigma = 0.25$ | **91.2** | **79.3** | **65.5** | 48.7 | 0.0 |
| $\sigma = 0.5$ | 81.5 | 67.0 | 56.1 | 45.3 | 35.5 |
| $\sigma = 1.0$ | 65.1 | 48.4 | 41.7 | 35.2 | 29.0 |

(d) Finetuned ViT

Table 6: Certified accuracy of four different classifiers on CIFAR-10 at varying levels of Gaussian noise $\sigma$, all using the same diffusion model.

| | Certified Accuracy at $\varepsilon$ (%) | | | | | |
|---|---|---|---|---|---|---|
| Noise | 0.0 | 0.5 | 1.0 | 1.5 | 2.0 | 3.0 |
| $\sigma = 0.25$ | **82.8** | **71.1** | 0.0 | 0.0 | 0.0 | 0.0 |
| $\sigma = 0.5$ | 77.1 | 67.8 | **54.3** | **38.1** | 0.0 | 0.0 |
| $\sigma = 1.0$ | 60.0 | 50.0 | 42.0 | 35.5 | **29.5** | **13.1** |

Table 7: Certified accuracy on ImageNet for varying levels of Gaussian noise $\sigma$.

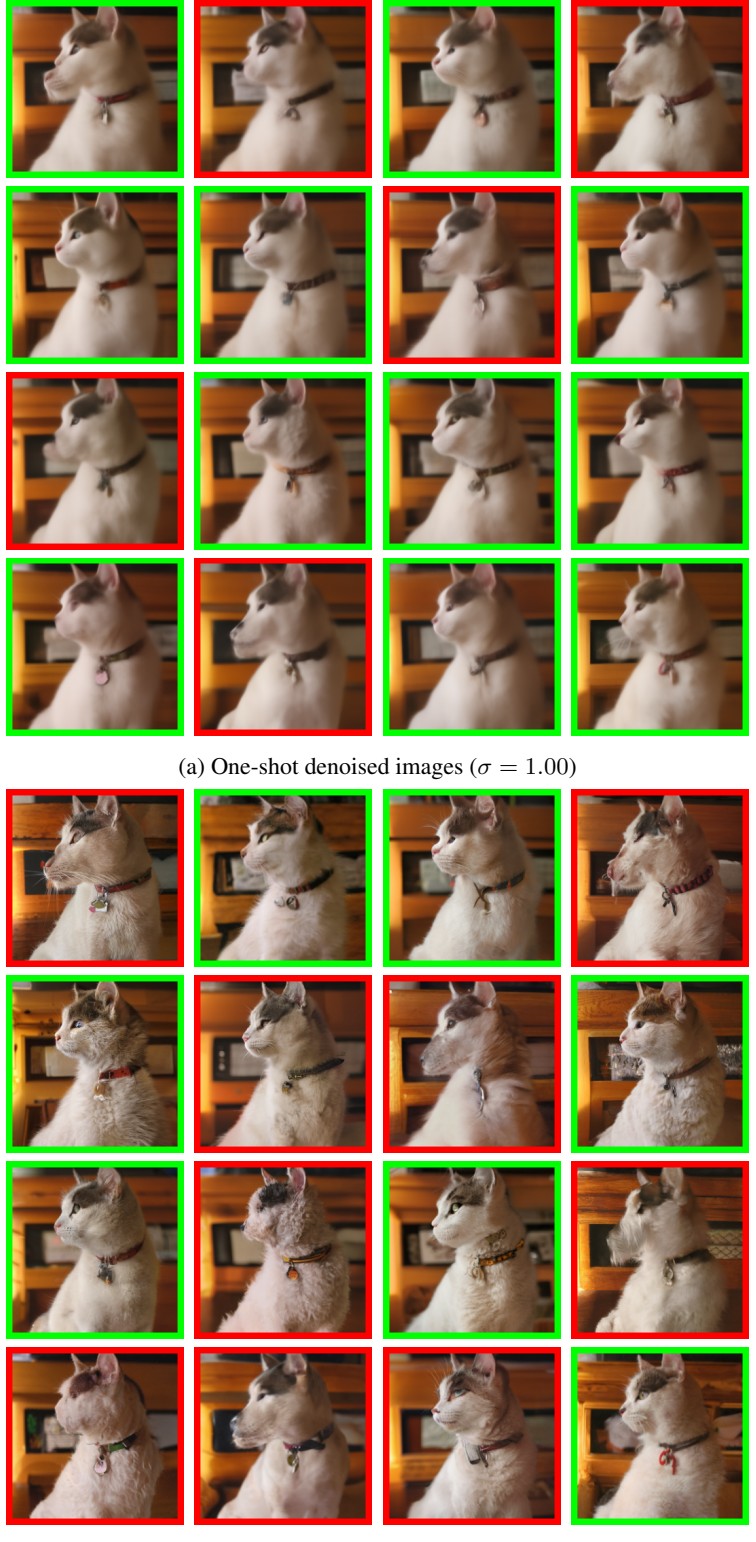

(a) One-shot denoised images ($\sigma = 1.00$)

(b) Multi-step denoised images ($\sigma = 1.00$)

Figure 4: Qualitative comparison of one-shot denoising and multi-step denoising. We show denoised images under random Gaussian noise ($\sigma = 1.00$). A green border is applied when the denoised images are correctly classified while a red border means that the classifier misclassifies the image.

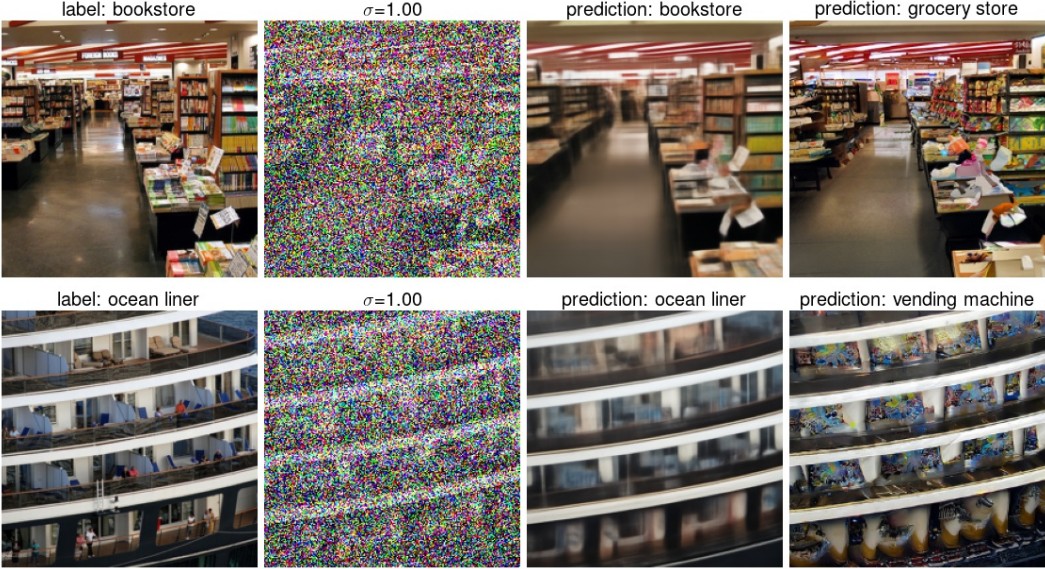

Figure 5: Additional intuitive examples for why multi-step denoised images are less recognized by the classifier. From left to right: clean images, noisy images with $\sigma = 1.0$, one-step denoised images, multi-step denoised images. For the denoised images, we show the prediction by the pretrained BEiT model.

