# OpenReview forum: "(Certified!!) Adversarial Robustness for Free!"
_ICLR.cc/2023/Conference — ICLR 2023 poster_

### Official Review · Reviewer_Kvhz · 2022-10-14

**Confidence:** 4
**Correctness:** 3
**Technical Novelty And Significance:** 2
**Empirical Novelty And Significance:** 4
**Recommendation:** 8

**Clarity, Quality, Novelty And Reproducibility:**

The paper is written very clearly. The only thing I believe should be clarified better is the latency description in Section 4 ("is 1.5 seconds per image on an A100 GPU at a batch size of 32"). Presumably, this is not the time it takes to verify an image but just to classify it so make this crystal-clear to all readers.
I have no negative comments about quality, novelty or reproducibility.

**Strength And Weaknesses:**

The method is technically very simple, and yet highly effective, which I view as a major strength. The result is significant because it is a "scale is all you need" type result for certified robustness, which has the potential to have a big impact on this subfield as a whole.
The authors also provide some additional experiments as ablations, which I believe to be valuable, i.e. showing that fine-tuning can help and demonstrating that a variety of noise levels improve the denoisers over more restricted noise levels.
The authors compare to many baselines which is good.

However, one point of criticism is that the authors do not actually provide any results on full diffusion as opposed to one-shot denoising. They provide convincing intuitive explanations of why one would expect that one-shot denoising is better suited for this task, but the examples they provide might be quite cherry-picked. I suspect that the reason why they do not provide numbers for comparison is that, as they admit, it is far too expensive to run this many steps on every single forward pass (of which many thousands are needed for randomized smoothing). I think the authors need to either a) include numbers for full diffusion (preferred option) or b) weaken their language in Section 3 where they claim in the paragraph on "High accuracy" that "Section 5 experimentally validates this hypothesis."

Another minor criticism is about the misleading title. I understand that this title is good for calling attention to this paper but I find "for free" to be very misleading, given that of course the accuracy vs. (certified) robustness tradeoff still persists and, crucially, that each verified sample still takes orders of magnitude more resources than a single prediction. A less snappy, though more accurate title would be much better.

**Summary Of The Paper:**

The paper demonstrates that one can achieve state-of-the-art certified $l_2$-robustness using the existing method of denoised smoothing, by simply replacing the custom denoiser by an off-the-shelf diffusion model.

**Summary Of The Review:**

I believe this paper is well-written and has significant results, especially because of its simplicity. I argue that this paper should be accepted at ICLR, under some minor revisions that I describe in my main review.

---

> ### Author Response · Authors · 2022-11-17
> **Response**
>
> We thank the reviewer for their positive assessment of our paper and their constructive comments.
>
> - **the authors do not actually provide any results on full diffusion as opposed to one-shot denoising**
>
> We will include more details in the final version of the paper, but with full diffusion steps on CIFAR-10 we see accuracy drop from 76.7% certified accuracy to 74% +/- 1% at eps=0.25, and from 63.0% at eps=0.5 to 61% +/- 1%. The margin of error here is sufficiently small to draw a statistically significant conclusion (p < .05), but in the final paper we will perform more experiments to reduce the margin of error further, along with performing experiments on ImageNet which are more expensive.
>
> - **I find "for free" to be very misleading, given that of course the accuracy vs. (certified) robustness tradeoff still persists**
>
> We apologize for this confusion. Our play on the “for free” term here is inspired by the “Adversarial Training for Free!” paper (https://arxiv.org/abs/1904.12843) which showed how to get empirical robustness with minimal overhead over standard training.
> Here, we show an analogous result for certified robustness, since we get robustness with off-the-shelf models.
>
> - **this is not the time it takes to verify an image but just to classify it**
>
> Yes, that is correct. We will clarify this in the paper. We believe this number is the more important latency number, since verification is an “off-line” process that the model developer would perform once to certify the model’s robustness for a given data distribution.

---

> > ### Comment · Reviewer_Kvhz · 2022-11-21
> > **Title still misleading**
> >
> > I acknowledge the authors' rebuttal and appreciate their additional preliminary results.
> >
> > Despite this, I still object to their title. I understand the authors' motivation for the title but I would still argue that the paper should be named more appropriately.
> > Note that the paper's results are also not fully analogous to the “Adversarial Training for Free!”-paper's results as hinted at in the rebuttal, since in that case at least the inference time did not change, which here it drastically does.

---

### Official Review · Reviewer_VkUH · 2022-10-25

**Confidence:** 4
**Correctness:** 3
**Technical Novelty And Significance:** 3
**Empirical Novelty And Significance:** Not applicable
**Recommendation:** 6

**Clarity, Quality, Novelty And Reproducibility:**

Clarity: very well organized
Quality: Provide new insights with an in-depth discussion
Novelty: There is no new contribution to the methodology of denoised randomized smoothing. But I think the overall insights and findings are sufficiently novel.
Reproducibility: Code is not provided; can't not verify reproducibility

**Strength And Weaknesses:**

Strength: The result shows the benefit of using the off-the-shelf diffusion model as the denoiser for improving certified robustness. Some limitations (e.g., fast degraded certified accuracy with increased perturbation levels) and some ablation studies in terms of one-step and multi-step denoising are discussed.

Weakness: There is no new contribution to the methodology of denoised randomized smoothing. My rating is based on the reported numerical improvement and the discussion.

**Summary Of The Paper:**

Based on the framework of denoised randomized smoothing (Salman et al. (2020)), the authors applied larger diffusion models as the denoiser and observed significantly improved certified robustness (when the perturbation range is not too larger) compared to existing methods.

**Summary Of The Review:**

In general, the claims are well supported by the presented results, and both advantages and limitations are discussed.

To understand whether the improved certified accuracy comes from the data itself, I would like to see an ablation study that uses the one-shot denoised images from diffusion models as data inputs, and then apply them to standard randomized smoothing. If there are performance gains compared to unperturbed images, then this result will show the improvement is due to the fact that denoised images are more robust to Gaussian perturbations than clean images.

---

> ### Author Response · Authors · 2022-11-17
> **Response**
>
> We thank the reviewer for their positive assessment of our paper and their constructive comments.
>
> - **There is no new contribution to the methodology of denoised randomized smoothing**
>
> We agree! We see this as a positive aspect of our paper. While prior works have mainly focused on improving the methodology of randomized/denoised smoothing (e.g., by combining it with adversarial training, or ensembling, etc.), we show that instantiating the existing methodology with state-of-the-art models provides much larger benefits.
>
> - **Code is not provided**
>
> We commit to releasing source code along with the final version of this paper.

---

### Official Review · Reviewer_PB4j · 2022-10-25

**Confidence:** 4
**Correctness:** 4
**Technical Novelty And Significance:** 3
**Empirical Novelty And Significance:** 3
**Recommendation:** 8

**Clarity, Quality, Novelty And Reproducibility:**

- The writing is overall clear and in good quality.
- Reproduction details are given yet no code is given.
- The work seems reasonably novel to me. It's a smart application of diffusion models in the certification robustness field.

**Strength And Weaknesses:**

Strength:
- An elegant solution to a challenging problem. The intuition and method are both relatively simple, yet highly effective. Although it is not so novel as the idea of using a denoiser is already proposed, using a diffusion model is novel, and pre-trained diffusion models significantly improve denoising accuracy.
- The one-step denoising is well-thought-of. It reduces computation costs by a great deal. Yet I'm curious why the certification accuracy is worse when the denoising result is more accurate.

Weakness:
- I hope to see more details on the hyperparameters for the diffusion process, including the noise scale during diffusion \alpha_t, and the number of diffusion steps chosen during smoothing as it is related to the smoothing level.
- I would like to see details on the backbone model for baseline models, as using ViT VS. ResNet could be very different in terms of clean accuracy on ImageNet.

**Summary Of The Paper:**

This paper proposes a method for certifying the adversarial robustness of off-the-shelf models with pre-trained Denoising Diffusion Probabilistic Model. The main idea is to use a denoiser after introducing noise for robustness certification and well-trained diffusion models can work well as denoisers. The noise introduced by the diffusion models can be used seen as the smoothing process and single-step denoising is then applied. Their main contribution is extending the denoising-based certification model to using a diffusion model. They take advantage of the recent advances in diffusion models and greatly improve the quality of denoising and certification. The experiments on ImageNet and CIFAR both show significant improvement in certification accuracy.

**Summary Of The Review:**

I lean to accept this paper as it wins me over with a simple yet effective method.

---

> ### Author Response · Authors · 2022-11-17
> **Response**
>
> We thank the reviewer for their positive assessment of our paper and their constructive comments.
>
> - **I'm curious why the certification accuracy is worse when the denoising result is more accurate**
>
> We provide some intuition for this in Section 5.1., where we show that multi-step diffusion can lead to reconstructed images with “hallucinated” high frequency components. As a result, these reconstructed images can be harder to classify correctly.
>
> - **more details on the hyperparameters for the diffusion process**
>
> *Could the reviewer clarify which hyperparameters are missing?*
> The noise scale \alpha_t is computed according to equation (4) from the noise deviation used for randomized smoothing.
> The number of diffusion steps is fixed to one in all our experiments (except for the ablation in Section 5.1).
>
> - **details on the backbone model for baseline models**
>
> We give details on our classifiers for CIFAR-10 and ImageNet (including their clean accuracy) at the beginning of Section 4.
> We also include clean accuracies of our robust model (in parenthesis) in tables 1-2-3.
> *Are there additional details that we can provide to help the reader?*

---

### Official Review · Reviewer_kRsg · 2022-10-26

**Confidence:** 4
**Correctness:** 3
**Technical Novelty And Significance:** 2
**Empirical Novelty And Significance:** 4
**Recommendation:** 6

**Clarity, Quality, Novelty And Reproducibility:**

The paper is clearly written. The technical novelty can be questionable, as the method itself does not introduce new component over existing method of Denoised smoothing and Diffusion model.

**Strength And Weaknesses:**

**Strength**

* The paper is easy-to-follow, and proposes a simple, easy-to-use method
* The paper presents an extensive evaluation covering large-scale dataset such as ImageNet, as well as exploring diverse architectures such as ViT
* The practice established in the paper could be useful in the future works given its significant improvements compared to previous similar attempts.


**Weakness**

* Although the paper claims a good practical performances by utilizing pre-trained models, but lacks on validating the effectiveness of the methodology itself: here, I could see the method can be also applied in the standard, non-off-the-shelf setup by training each diffusion model and classifier on the target dataset, e.g., CIFAR-10 from scratch for a fairer comparison - which I think a good ablation study to add.
* The evaluation could also be strengthen by comparing Average Certified Radius (ACR) of the models which is often in the literature for a metric that considers both accuracy and robustness.
* A discussion on the cost for certification (in other words, inference cost/overhead) compared to existing model would be helpful and worth to be added.



**Summary Of The Paper:**

The paper observes that a state-of-the-art Diffusion models could be a good denoiser applicable for Denoised smoothing when combined with pre-trained classifiers, specifically to construct an off-the-shelf smoothed classifier that offers robustness certificates. The paper establishes several practices for this pipeline which enables a significant improvements in performance compared to previous similar attempts, e.g., the use of one-shot denoising unlike standard practice, and a proper way to scale the given noisy image to make the pipeline in compliance to the certification protocol.

**Summary Of The Review:**

Despite its lack of technical novelty, I think the empirical novelty of the observation and its practical significance could overweigh it thus for now I lean to accept for the paper.

---

> ### Author Response · Authors · 2022-11-17
> **Response**
>
> We thank the reviewer for their positive assessment of our paper and their constructive comments.
>
> - **the method can be also applied in the standard, non-off-the-shelf setup**
>
> Yes, we agree and this is exactly the point of our ablation study on CIFAR-10 (bottom of page 6, and Table 3). The diffusion model we use was trained solely on CIFAR-10. We find that using a classifier trained solely on CIFAR-10 (Wide ResNet) reduces robust accuracy by about 5-6 percentage points. If we further fine-tune this classifier on denoised data (again, using only CIFAR-10) then we beat the prior state-of-the-art while using only CIFAR-10 data.
>
> - **A discussion on the cost for certification compared to existing model would be helpful**
>
> We do include throughput numbers for our models on CIFAR-10 and ImageNet in section 4. Compared to most prior approaches that use a stand-alone classifier at inference time, our method incurs the additional overhead of running the diffusion model (for one-shot denoising).

---

### Public Comment · ~Linyi_Li1 · 2022-11-06
**More baselines can be included**

Dear authors,

It is a great work that shows large diffusion models can work as denoisers to bring certified robustness! Thanks for all the efforts. Maybe this paper can be benefited from comparing with one published baseline:

[1] (our DRT framework for certified training) Yang, Zhuolin, et al. "On the certified robustness for ensemble models and beyond." ICLR 2022.

Furthermore, the robustness guarantees may be converted and thus compared with strong baselines against $\ell_\infty$-bounded attacks [2,3]:

[2] Zhang, Bohang, et al. "Boosting the certified robustness of l-infinity distance nets." ICLR 2022.

[3] Shi, Zhouxing, et al. "Fast certified robust training with short warmup." NeurIPS 2021.

---

> ### Author Response · Authors · 2022-11-17
> **Added DRT Baseline**
>
> Hi, thanks for your comment.
> Yes, we have included a comparison with [1] in the paper. The results is as follows:
>
> - **CIFAR-10**
> | 0.25 | 0.5 | 0.75 | 1.0 |
> |------|-----|-------|-----|
> |$^{(81.5)}70.4$ | $^{(72.6)}60.2$ |  $^{(72.6)}50.5$ | $^{(72.6)}39.5$|
>
>
> - **ImageNet**
> | 0.5 | 1.0 | 1.5 | 2.0 | 3.0 |
> |------|-----|-------|-----|------|
> |$^{(68.0)}57.0$ | $^{(55.2)}44.4$ | $^{(49.8)}39.8$ | $^{(49.8)}30.4$ | $^{(49.8)}23.4$ |
>
>
> Regarding $\ell_\infty$-robustness, we could indeed convert our $\ell_2$ certificates into $\ell_\infty$ certificates in a naive way (simply by dividing our certified $\ell_2$ radius by $\sqrt{d}$).
> This leads to better $\ell_\infty$ certificates than prior work, for low epsilons.
>
> For example, on CIFAR-10 we achieve 65.5% robust accuracy for $\ell_2$ perturbations of size 0.5.
> This corresponds to $\ell_\infty$ perturbations of size at least $0.5/\sqrt{32\cdot 32\cdot 3} \approx 2.3/255$.
> In contrast, the prior state-of-the-art robust accuracy for perturbations of size 2/255 appears to be 62.6% from https://arxiv.org/abs/2002.03517 (this result was also obtained by converting $\ell_2$ certificates into $\ell_\infty$ certificates).

---

> > ### Public Comment · ~Linyi_Li1 · 2022-11-18
> > **Thanks for your reply**
> >
> > Thanks for updating the paper and taking these comments into account. As a note, in https://arxiv.org/abs/1906.04584, Table 3 seems to provide another baseline on CIFAR-10 against $2/255$ $\ell_\infty$ perturbations.
> >
> > Good luck with the submission!

---

### Public Comment · ~Yijiang_Pang1 · 2022-11-09
**terminologies**

Dear authors,
It is a very good work. Since the score-based input purification models can practically induce adv-robustness also mentioned by the paper, the boundaries become blurred. But, it may help us understood the work better if the differences of some terminologies can be emphasized. Such as,
probabilistic certified robustness VS (general) certified robustness,
adversarial robustness under random noise VS (worst-case) adversarial robustness under adv-attack.

---

### Comment · Area_Chair_KpWA · 2022-11-15
**Please engage before the author-reviewer discussion closes**

Dear authors and reviewers,

The first phase of the discussion period is about to close on November 18.

For authors, please make sure to submit your rebuttal by the deadline. Leave some time for the reviewers to read it and respond while you are still allowed to further engage with them. Interactions between authors and reviewers are very important for the quality of the review process, so please make sure to engage.

For reviewers, please try to acknowledge and respond to the authors' rebuttal while the discussion period is still open for them to further interact with you.

Thank you for your participation in the review process!

Best,
The AC

---

### Decision · Program_Chairs · 2023-01-20

**Decision:**

Accept: poster

**Justification For Why Not Higher Score:**

The execution is solid and the results show the method is highly effective. However, the contribution is also incremental.

**Justification For Why Not Lower Score:**

All reviewers are positive, and all minor concerns were addressed.

**Metareview: Summary, Strengths And Weaknesses:**

All four reviewers recommend accepting the paper (6-8-6-8). The paper proposes using a state-of-the-art diffusion model as a denoiser for Denoised Smoothing, resulting in a simple and yet highly effective solution for certifiably robust pre-trained classifiers. The experimental validation is strong. Minor concerns raised by reviewers were answered in the rebuttal.

Without questioning the validity of the method, the title is considered to be slightly misleading. The authors are encouraged to consider a more accurate title for the paper.

**Note From Pc:**

if the above contains the word "oral" or "spotlight" please see: "oral" presentation means -> notable-top-5% and "spotlight" means -> notable-top-25%. As stated in our emails, we are disassociating presentation type from AC recommendations